# The effect of comprehensive intervention for childhood obesity on dietary diversity among younger children: Evidence from a school-based randomized controlled trial in China

Haiquan Xu[1], Olivier Ecker[2], Qian Zhang[3], Songming Du[4], Ailing Liu[3], Yanping Li[5], Xiaoqi Hu[3], Tingyu Li[6], Hongwei Guo[7], Ying Li[8], Guifa Xu[9], Weijia Liu[10], Jun Ma[11], Junmao Sun[1], Kevin Chen[2,12]*, Guansheng Ma[13]*

1 Institute of Food and Nutrition Development, Ministry of Agriculture and Rural Affairs, Beijing, China, 2 International Food Policy Research Institute, Washington, DC, United States of America, 3 National Institute for Nutrition and Health, Chinese Center for Disease Control and Prevention, Beijing, China, 4 Chinese Nutrition Society, Beijing, China, 5 Department of Nutrition, Harvard T. H. Chan School of Public Health, Boston, Massachusetts, United States of America, 6 Chongqing Children's Hospital, Chongqing, China, 7 School of Public Health, Fudan University, Shanghai, China, 8 Harbin Medical University, Harbin, China, 9 Department of Public Health, Shandong University, Jinan, China, 10 Guangzhou Center for Disease Control and Prevention, Guangzhou, China, 11 Institute of Child and Adolescent Health, School of Public Health, Peking University, Beijing, China, 12 China Academy for Rural Development, Zhejiang University, Hangzhou, China, 13 Department of Nutrition and Food Hygiene, School of Public Health, Peking University, Beijing, China

* K.Chen@cgiar.org (KC); mags@bjmu.edu.cn (GM)

**Data Availability Statement:** All relevant data are within the paper and its Supporting Information files.

## Abstract

### Background

Little evidence from developing countries on dietary transition demonstrates the effects of comprehensive childhood obesity interventions on dietary diversity and food variety among younger children. This study aimed to evaluate the effects of comprehensive childhood obesity interventions on dietary diversity among younger children.

### Methods

A total of 4846 children aged 7–13 years were included based on a multicenter randomized controlled trial for childhood obesity interventions in 38 primary schools. Nutrition education intervention (NE), physical activity intervention (PA) and comprehensive intervention including both NE and PA (CNP) were carried out separately for 2 semesters. Dietary Diversity Score (DDS9 and DDS28 for 9 and 28 food groupings, respectively), Food Variety Score (FVS, the number of food items) and the proportions of different foods consumed were calculated according to the food intake records collected with the 24-h dietary recall method.

### Results

The intervention effects per day of comprehensive intervention group were 0 (95% Confidence Interval (CI): 0, 0.1; p = 0.382) on DDS9, 0.1 (95% CI: -0.1, 0.2; p = 0.374) on DDS28 and 0.1 (95% CI: -0.1, 0.3; p = 0.186) on FVS of overall diet, which was 0.1 (95% CI: 0, 0.1;

**Funding:** This study was funded by China Ministry of Science & Technology [2008BAI58B05], Chinese National Natural Science Foundation Project [71804079] and Science and Technology Innovation Project of the Chinese Academy of Agricultural Sciences [CAAS-ASTIP-2019-IFND]. The funders had no role in study design, data collection and analysis, decision to publish, or preparation of the manuscript.

**Competing interests:** The authors have declared that no competing interests exist.

p < 0.001) on DDS9, 0 (95% CI: 0, 0.1; p = 0.168) on DDS28 and 0.1 (95% CI: 0, 0.1; p = 0.067) on FVS of dietary scores of breakfast only. Additionally, CNP group had greater increases in cereals, meat and fruits, and more decreases in eggs, fish and dried legumes consumption proportions as compared with the control group. Decreasing side effect on dietary diversity and food variety were found for PA intervention, but not for NE intervention only.

## Conclusions

Though the comprehensive obesity intervention didn't improve the overall dietary diversity per day, the positive intervention effects were observed on breakfast foods and some foods' consumption.

## Introduction

With the economic development in China, the increase in unhealthy body weight has led to a fast increase in obesity prevalence among children. The Chinese Residents' Nutrition and Chronic Disease Report (2015) showed that the prevalence of overweight and obesity among children aged 6–17 years increased from 6.6% in 2002 to 16.0% in 2012 [1]. To control childhood obesity, school-based intervention programs focusing on nutrition education, physical activity or both have increasingly emerged as important strategies in China, mainly focused on shaping healthy eating habits and balancing energy intake and expenditure [2].

Dietary diversity, representing the consumption of various food items within and between food groups, is a strong predictor of dietary quality, defined as micronutrient adequacy, in developing countries [3]. Previous studies revealed that dietary diversity is associated with the micronutrient adequacy of diets and anthropometry in children [3, 4]. According to the dietary guidelines in many countries, dietary diversity is one of the characteristics of a healthy diet [5, 6]. By testing different food groupings, food groups (ranging from 7 to 21 groups) and accuracy of indicators, dietary diversity score (DDS) and food variety score (FVS) have been widely used as effective indicators [7, 8]. In addition to a positive relationship between DDS and nutrient intake being reported, the inverse association between DDS and chronic diseases [9–11] have been revealed by several studies.

Research has also revealed the dietary diversity associated with high energy intake [12]. Nutrition education is viewed as a key strategy for promoting healthy eating habits, while a physical activity-friendly school environment is also associated with a lower risk of obesity [13]. Many researchers found that diet quality could increase among children after nutrition education intervention, and some studies indicate that a lifestyle intervention plus nutrition education could improve dietary diversity [14, 15]. One multicenter randomized controlled trial of a comprehensive school-based intervention study focusing on childhood obesity was implemented in China [16]. The evaluation indicated that comprehensive intervention was more effective than only nutrition education or only physical activity on childhood obesity prevention [17, 18]. And we observed a moderately significant effect on the combined prevalence of overweight and obesity, which increased by 1.5 percent in the control group and 0.2 percent in the CNP group after intervention. The effect was significantly stronger among girls than boys (-1.4% vs. -0.9%). However, we did not find a significant effect in the nutrition education group or the physical activity group. Several school-based interventions for childhood

obesity have revealed a positive effect on changing eating behaviors and improving vegetable and fruit consumption and the availability of healthy foods, but few have focused on dietary diversity or food variety [19–21]. Though research showed that school children with a high level of physical activity presented a better quality of the diet [22], we wanted to know whether childhood obesity intervention will increase dietary diversity. To our knowledge, little previous research has evaluated the effects of childhood obesity interventions on dietary diversity among younger children in developing countries undergoing dietary transition. Therefore, the objective of this study was to explore the effects of nutrition education and physical activity as childhood obesity interventions on dietary diversity among younger children in China.

## Materials and methods

### Study design and sample size

This trial was designed into two parts. One study was a cluster-randomized controlled trial for nutrition education intervention (NE) and physical activity intervention (PA) separately in Beijing. The other one was a multicenter cluster-randomized controlled trial for a comprehensive intervention including nutrition education and physical activity (CNP) in 5 centers, in Shanghai, Chongqing, Guangzhou, Jinan and Harbin. Multi-step randomized cluster sampling method was used for subjects' selection. First, 1–3 districts were selected randomly in the lottery in each city, then the schools were selected randomly in the lottery in each selected district. The schools were selected according to some inclusion criteria: 1) non-boarding school; 2) the prevalence of obesity, based on the routine physical examination records, was above 10%; 3) Providing school lunch feeding, and more than 50% students have lunch at school. Eight schools selected randomly were divided into three groups (3 schools for NE, 3 schools for PA and 2 schools as a shared control) in Beijing, and six schools from each other city were randomly sorted into 2 groups (3 for CNP and 3 for control). In total, 38 primary schools were included in this trial. Two classes were selected from each grade (1st to 5th) in every school, but only grades 2 to 4 were designed to collect the dietary records. The program was implemented for 2 semesters from May 2009 to May 2010. NE was also targeted towards parents, teachers, and health workers in treated schools. The detailed design information has been described in previous articles [16, 18].

This trial was approved by the Ethical Review Committee of the National Institute for Nutrition and Food Safety, Chinese Center for Disease Control and Prevention. The informed consent was voluntarily signed by the participants' parents or their guardians. The trial was registered at the Chinese Clinical Trial Register (number ChiCTR-PRC-09000402).

The present analysis on dietary diversity was not the primary outcome for this trial. The calculation of sample size was performed according to the changes of DDS9 in the CNP. The interclass correlation coefficient for DDS9 was 0.18. with a sample size of 4051, we will have 90% power to detective an effect size of as much as 0.6 from 30 schools located in 5 centers, at double side 0.05 level. Our effect size was 0.61, with sample size of 4069, stronger than the minimum detectable level. Only the children in 2nd grade or higher had their dietary records collected, 4933 (both in 5 centers and in Beijing) children with dietary records were extracted from the total participants. Finally, 4846 subjects' information with both baseline and ending dietary records was used. The trial profile for data analysis is shown in Fig 1.

### Intervention implementation

Table 1 shows the intervention measures for different intervention groups. In NE intervention schools, one nutrition education handbook was developed [23] for the students, courses on nutrition and health for the students, parents, teachers and health workers were designed

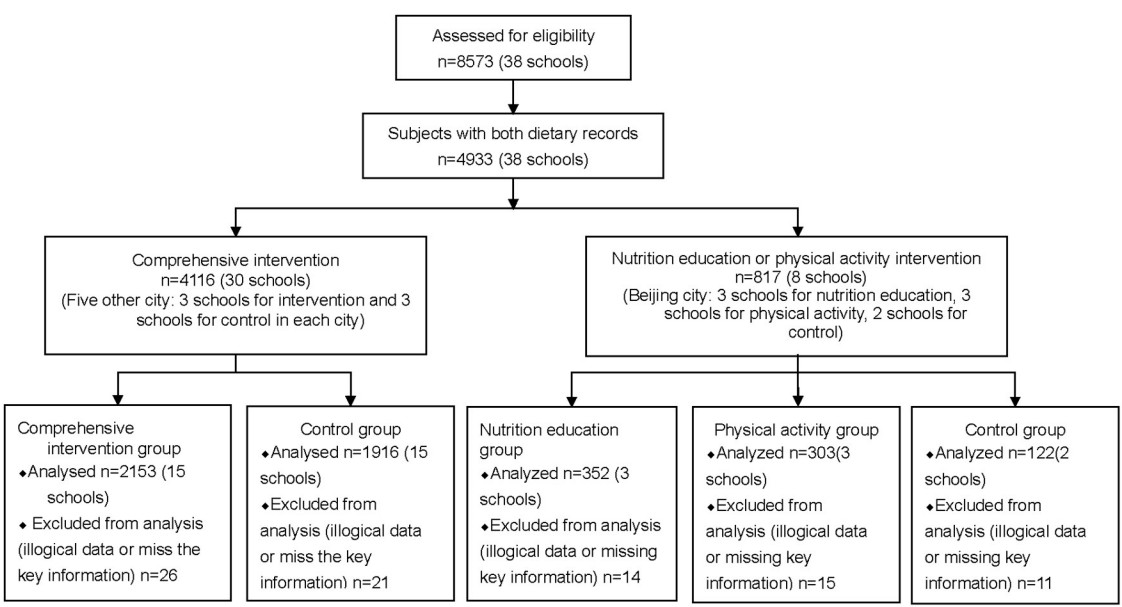

**Fig 1. The trial profile for data analysis.**

separately. Furthermore, "Dietary Pagoda for Chinese people" posters were displayed in the classrooms for NE intervention groups. In PA intervention schools, the "Happy 10", which was a classroom-based physical activity program for primary school students was used [18, 24]. Furthermore, students, parents, health workers and teachers received the PA education from the program. To improve the home environment, the parents in the intervention groups were also involved, including but not limited to sending them physical activity education bulletins. In CNP schools, all above interventions including both NE and PA were implemented. In order to ensure the normal operation of the intervention, the supervisors from project office went to each center to carry out project supervision during the program period. The process evaluation indicated that the school carried out as proposal. The physical activity level was measured with energy monitor among 868 students (304 in CNP group and 301 in control group in 5 cities, 111 in PA group and 112 in NE group in Beijing). The energy expenditure

**Table 1. The interventions for different intervention groups.**

| Intervention groups | Interventions | Sites |
|---|---|---|
| Nutrition education (NE) | 1. The nutrition handbook was distributed to students. | Beijing |
| | 2. Courses on nutrition and health were given 6 times to students, 2 times to parents and 4 times to teachers and health workers. The content included proportion of three meals, how to choose healthy food, reducing eating out, unhealthy fast food sugar sweetened beverage and snacks. | |
| | 3. Displaying poster of "Dietary Pagoda for Chinese people". | |
| | 4. Two class meetings for nutrition and health. | |
| Physical activity (PA) | 1. Course on physical activity given to parents | Beijing |
| | 2. Twenty-minute "Happy 10" physical activity for students on each school day | |
| | 3. Two class meetings for physical activity and health. | |
| Comprehensive intervention including nutrition education and physical activity (CNP) | All the above interventions including both NE and PA were implemented. | The other 5 cities (Jinan, Chongqing, Harbin, Guangzhou and Shanghai) |

increased higher in CNP group (77.0 kcal/day) than in control group (63.5 kcal/day, p = 0.967) in five cities study, higher in PA group (96.4 kcal/day) than in NE group (76.7 kcal/day, p = 0.408) in Beijing study.

## Anthropometric measurements

The physical examination was carried out in school. Height was measured to an accuracy of 1 mm with a free-standing stadiometer mounted on a rigid tripod (GMCS-I, Xindong Huateng Sports Equipment Co. Ltd., Beijing, China). One overnight fasting body weight was measured to the nearest 0.1 kg on a digital scale (RGT-140, Wujin Hengqi Co. Ltd., Changzhou, China). Body mass index (BMI) was calculated as weight in kilograms divided by height in meters squared (kg/m$^2$). Overweight, obesity and thinness were defined per the China national standard for screening for overweight and obesity among school-age children and adolescents with BMI at baseline [25].

## Sociodemographic information

Sociodemographic information was collected with a parent questionnaire. Parental education level was defined by maternal education level when available, which was supplemented by paternal education level as a surrogate when maternal education level was not available. The family's income level was classified as household income per capita monthly in 2009.

## Dietary diversity and food variety measurements

Information on dietary intake was collected with a 24-h dietary recall method for three consecutive days (two weekdays and one weekend day); the records were administered by the subjects themselves or with their parents' help. The children were taught how to fill out the dietary records by the trained research assistants. DDS and FVS per day were used as primary outcome indicators for assessing dietary diversity and food variety. Literature reveals that the quality of breakfast is more important [26, 27] than the others, so the DDS and FVS per meal (breakfast, lunch and supper) were calculated separately as the second outcome indicators.

DDS9 was calculated based on 9 food groups' consumption. The food groups were categorized according to the FAO protocol in 2013 as Starchy staples; Dark-green leafy vegetables; Other vitamin A-rich fruits and vegetables; Other fruits and vegetables; Organ meat; Meat and fish; Eggs; Legumes, nuts and seeds; and Milk and milk products [28]. One score was assigned if at least 1 food from a given food group was consumed and 0 if not. First, the score for each day was calculated according to the assignment method, and then the average of three-day DDS was taken as the subject's final score. FVS was calculated by counting all the food items recorded in dietary records with the same method as DDS. The different food items were defined according to China Food Composition Tables 2004.

DDS28, as another food grouping method based on 28 food groups, was referred to confirm the robustness of the evaluation of dietary diversity. The 28 food groups were categorized as follows: Wheat; Fried wheat; Rice; Refined grains; Other cereals; Starch tubers; Beans; Soybean products; Nuts; Deep-color vegetables (carotene content $\geq$ 500 μg/100 g); Light-color vegetables (carotene content < 500 μg/100 g); Pickled vegetables; Edible fungi and algae; Fruits; Pork; Poultry; Beef/lamb/other red meat; Seafood; Organ meat; Processed meat; Eggs; Milk powder and cheese; Milk and yogurt; Catsup; Beverages; Sugar; Fast food; and Cakes [29].

## Proportions of different foods' consumption

To observe the effects of different interventions on main food consumption, we analyzed the changes in the proportion of subjects consuming some specific foods over 3 days. The foods were categorized according to the China Food Composition Tables 2004 as Cereals ($>3$ times /3 days); Meat ($>3$ times /3 days); Vegetables ($>3$ times /3 days); Fruits ($\geq 1$ time /3 days); Dairy ($\geq 1$ time /3 days); Eggs ($\geq 1$ time /3 days); Fish and shellfish ($\geq 1$ time /3 days); Fungi and algae ($\geq 1$ time /3 days); Dried legumes ($\geq 1$ time /3 days); Nuts and seeds ($\geq 1$ time /3 days); Snack ($\geq 1$ time /3 days); Fast foods ($\geq 1$ time /3 days); Beverages ($\geq 1$ time /3 days); Sugars and preserves ($\geq 1$ time /3 days).

## Statistical analysis

The continuous variables are expressed as mean and standard deviation. The intra-cluster correlation coefficient was 0.18, 0.13 and 0.13 for DDS9, DDS28 and FVS in CNP study respectively. The liner regression model was used for the estimation of the intervention effect on DDS, FVS and the proportion of children consuming different food groups. A linear mixed model adjusting for confounding factors (weight status, sex, age, parental education level and family income level) was used to compare the changes in continuous variables between the intervention groups and control group. The proportions were compared with the generalized linear mixed model (GLMM). The school was used as a random-effect variable. The fixed-effect variables included sex, age, parental education level, family income level and intervention type. The statistical significance level was set at $P < 0.05$ for overall and $P < 0.05/n$ for subgroup (n represented the number of subgroups). The SAS software package version 9.2 (SAS Institute Inc., Cary, NC) was used for analysis.

The treatment effects were estimated using Eq (1) for DDS and FVS and Eq (2) for proportions of children consuming certain foods.

$$Y_{ij} = \gamma_{00} + \beta_0 \text{Cluster} + \beta_{01} \text{Treat} + \beta_1 X_{1ij} + \beta_P X_{Pij} + \varepsilon_{ij} \tag{1}$$

$$\text{Logit}(P_{ij}) = \gamma_{00} + \beta_0 \text{Cluster} + \beta_{01} \text{Treat} + \beta_1 X_{1ij} + \beta_P X_{Pij} + \varepsilon_{ij} \tag{2}$$

In Eq (1), $Y_{ij}$ is the outcome of change from baseline to end for child i at level j; $\gamma_{00}$ is the intercept parameter; Cluster is random effect indicator; $\beta_{01}$ reveals the intervention effect; Treat is an indicator that equals 1 if the student is in the treatment arm and 0 otherwise; $X_{1ij}$ is the outcome at baseline and $X_{Pij}$ is a vector of control variables at baseline. The same linear Eq (2) was applied to binary outcomes with a logit link as part of generalized linear models.

## Results

### Baseline characteristics

The sample size of children was 4846 in total (777 in Beijing and 4069 in the other five cities) in this analysis. Tables 2 and 3 present the characteristics of the subjects in comprehensive intervention study and single intervention study respectively, including age, sex, parental educational level and family economic level. Except for the significant economic and educational level differences between the CNP and its control group and the significant sex proportion difference among the NE group, PA group and their control groups, no other significant difference was found between the intervention and control groups in basic characteristics.

**Table 2. Characteristics of the subjects at baseline in comprehensive intervention study.**

| Characteristic | Control group | CNP group |
|---|---|---|
| Total, (N) | 1916 | 2153 |
| Age, (years, mean ± SD) | 9.0±1.2 | 9.0±1.2 |
| Sex, (N (%)) | | |
| Boys | 927 (48.4) | 1037 (48.2) |
| Girls | 989 (51.6) | 1116 (51.8) |
| BMI, (kg/m², mean ± SD) | 17.1±3.1 | 17.2±3.3 |
| Weight status, (N (%)) | | |
| Thinness | 261 (14.3) | 312 (16.2) |
| Normal weight | 1159 (63.4) | 1161 (60.1) |
| Overweight | 209 (11.4) | 245 (12.7) |
| Obesity | 200 (10.9) | 213 (11.0) |
| Parental educational level, (N (%)) | | |
| Low | 18 (1.0) | 21 (1.0) ** |
| Middle | 813 (43.9) | 787 (37.6) |
| High | 1023 (55.2) | 1287 (61.4) |
| Family's economic level (Yuan/month/per family member), N (%) | | |
| ≤1,500 | 830 (45.0) | 846 (40.8) * |
| 1,501–2,500 | 483 (26.2) | 591 (28.5) |
| >2,500 | 533 (28.9) | 638 (30.7) |

BMI, body mass index; CNP, comprehensive intervention including nutrition education and physical activity; NE, nutritional education intervention; PA, physical activity intervention.

The category of parental educational level: low: illiterate; middle: elementary or middle school; high: high school or above.

Mixed-effects model was used for comparison,

** $p < 0.01$,

* $p < 0.05$.

## Dietary diversity and food variety

The intervention effects on DDS and FVS according to dietary intake are shown in Table 4 for CNP study and in Table 5 for NE or PA study. Compared with the control group, the effects of CNP were 0 (95% Confidence Interval (CI): 0, 0.1; p = 0.382) on DDS9, 0.1 (95% CI: -0.1, 0.2; p = 0.374) on DDS28 and 0.1 (95% CI: -0.1, 0.3; p = 0.186) on FVS. No significant difference of intervention effects was found across subgroups. When we evaluated the dietary scores based on the breakfast foods only, positive intervention effects of CNP group were observed significant on DDS9 (0.1; 95% CI: 0, 0.1; p < 0.001), and nonsignificant on DDS28 (0; 95% CI: 0, 0.1; p = 0.168) and FVS (0.1; 95% CI: 0, 0.1; p = 0.067).

The effects on DDS9, DDS28 and FVS were 0.1 (95% CI: -0.1,0.3; p = 0.294), 0.1 (95% CI: -0.3, 0.4; p = 0.747) and 0.4 (95% CI: -0.2,0.9; p = 0.201) in the NE group and -0.1 (95% CI: -0.2, 0; p = 0.037), -0.3 (95% CI: -0.4,-0.1; p = 0.005) and -0.3 (95% CI: -0.6, 0; p = 0.021) in the PA group, respectively. No significant effects on DDS or FVS were found at breakfast in NE group or PA group.

## Food group consumption changes

The effects of the three interventions on food group consumption changes are shown in Tables 6 and 7. Compared with the control group, the proportions of children consuming cereals

**Table 3. Characteristics of the subjects at baseline in NE or PA intervention study.**

| Characteristic | Control group | NE group | PA group |
|---|---|---|---|
| Total, (N) | 122 | 352 | 303 |
| Age, (years, mean ± SD) | 9.1±1.4 | 9.4±1.3 | 9.0±1.3** |
| Sex, (N (%)) | | | |
| Boys | 70 (57.4) | 148 (42.0) | 141 (46.5) * |
| Girls | 52 (42.6) | 204 (58.0) | 162 (53.5) |
| BMI, (kg/m$^2$, mean ± SD) | 16.3±2.5 | 16.9±2.9 | 16.8±3.1 |
| Weight status, (N (%)) | | | |
| Thinness | 12 (9.8) | 31 (8.8) | 23 (7.6) |
| Normal weight | 93 (76.2) | 268 (76.1) | 224 (73.9) |
| Overweight | 10 (8.2) | 28 (8.0) | 27 (8.9) |
| Obesity | 7 (5.7) | 25 (7.1) | 29 (9.6) |
| Parental educational level, (N (%)) | | | |
| Low | 2 (1.7) | 10 (2.9) | 8 (2.7) |
| Middle | 84 (71.8) | 236 (69.2) | 190 (64.6) |
| High | 31 (26.5) | 95 (27.9) | 96 (32.7) |
| Family's economic level (Yuan/month/per family member), N (%) | | | |
| ≤1,500 | 62 (54.4) | 162 (48.2) | 157 (53.0) |
| 1,501–2,500 | 28 (24.6) | 92 (27.4) | 83 (28.0) |
| >2,500 | 24 (21.1) | 82 (24.4) | 56 (18.9) |

BMI, body mass index; CNP, comprehensive intervention including nutrition education and physical activity; NE, nutritional education intervention; PA, physical activity intervention.

The category of parental educational level: low: illiterate; middle: elementary or middle school; high: high school or above.

Mixed-effects model was used for comparison,

** $p < 0.01$,

* $p < 0.05$.

(> 3 times/3 days) and fruits (≥ 1 time/3 days) increased significantly, and Eggs (≥1 time /3 days), fish (≥1 time /3 days) and dried legumes (≥1 time /3 days) decreased significantly in the CNP group, with changes of 5.4 (-5.4 vs. -10.8 for treat vs. control; Odds Ratio (OR): 1.4 (95% CI: 1.1,1.7); p = 0.009) and 6.9 (5.0 vs. -1.9; OR: 1.4 (95% CI: 1.1,1.7); p = 0.003) percent, -4.3 (-0.1 vs. 4.2 for treat vs. control; OR: 0.8 (95% CI: 0.7,1.0); p = 0.031), -5.9 (-8.4 vs. -2.5; OR: 0.7 (95% CI: 0.6,0.9); p = 0.005) and -10.8 (-8.3 vs. 2.5; OR: 0.6 (95% CI: 0.5,0.7); p < 0.001) percent, respectively. The proportion changes among the NE group in the consumption of vegetables (> 3 time /3 days), fruits (≥ 1 time/3 days), Fungi and algae (≥1 time /3 days), dried legumes (≥ 1 time/3 days), snack (≥ 1 time /3 days) and fast foods (≥1 time / 3 days) were 20.2 (13.7 vs. -6.5 for treat vs. control; OR: 2.2 (95% CI: 1.2,4.5); p = 0.014), -26.7 (-20.2 vs. 6.5; OR: 0.3 (95% CI: 0.2,0.6); p = 0.001), 22.3 (25.6 vs. 3.3; OR: 2.5 (95% CI: 1.1,5.0); p = 0.024), -16.9 (2.0 vs. 18.9; OR: 0.5 (95% CI: 0.2,0.8); p = 0.011), -22.5 (-30.7 vs. -8.2; OR: 0.3 (95% CI: 0.1,0.7); p = 0.006) and -22.1 (-13.1 vs. 9.0; OR: 0.4 (95% CI: 0.2,0.7); p = 0.002) percent, respectively. The PA group showed no significant increases but significant decreases (or relative decreases) in the proportions of children consuming fruits (≥ 1 time/3 days), dried legumes (≥ 1 time/3 days) and fast foods (≥ 1 time/3 days), which were -16.8 (-10.3 vs. 6.5 for treat vs. control; OR: 0.5 (95% CI: 0.3,0.9); p = 0.023), -13.6 (5.3 vs. 18.9; OR: 0.6 (95% CI: 0.3,1.0); p = 0.048) and -16.3 (-7.3 vs. 9.0; OR: 0.5 (95% CI: 0.3,0.9); p = 0.027) percent compared with the control group, respectively.

**Table 4. The outcomes for intervention on DDS and FVS in CNP intervention study (mean ± SD).**

| Group | Variable | Control group | | CNP group | | | |
|---|---|---|---|---|---|---|---|
| | | Baseline | Change | Baseline | Change | Effect (95% CI) | P-value |
| **For one day** | | | | | | | |
| Total | DDS9 | 4.3±1.2 | -0.2±1.4 | 4.4±1.0 | -0.2±1.3 | 0 (0, 0.1) | 0.382 |
| | DDS28 | 6.7±2.2 | -0.5±2.6 | 6.7±1.9 | -0.5±2.3 | 0.1(-0.1, 0.2) | 0.374 |
| | FVS | 8.7±3.7 | -0.9±4.3 | 8.5±2.9 | -0.7±3.3 | 0.1 (-0.1, 0.3) | 0.186 |
| Boys | DDS9 | 4.2±1.2 | -0.2±1.4 | 4.3±1.0 | -0.3±1.3 | 0 (-0.1, 0.1) | 0.807 |
| | DDS28 | 6.4±2.1 | -0.6±2.5 | 6.5±1.9 | -0.6±2.3 | 0.1 (-0.1, 0.2) | 0.453 |
| | FVS | 8.2±3.5 | -1.0±4.1 | 8.2±2.9 | -0.9±3.3 | 0.1 (-0.1, 0.4) | 0.324 |
| Girls | DDS9 | 4.4±1.2 | -0.2±1.5 | 4.4±1.0 | -0.1±1.2 | 0 (0, 0.1) | 0.334 |
| | DDS28 | 6.9±2.2 | -0.5±2.7 | 6.8±1.9 | -0.4±2.2 | 0 (-0.1, 0.2) | 0.643 |
| | FVS | 9.1±3.8 | -0.9±4.5 | 8.8±2.9 | -0.6±3.3 | 0.1 (-0.1, 0.3) | 0.411 |
| Normal weight & thinness | DDS9 | 4.3±1.2 | -0.2±1.4 | 4.4±1.0 | -0.2±1.3 | 0.1 (-0.1, 0.2) | 0.488 |
| | DDS28 | 6.7±2.2 | -0.6±2.6 | 6.7±1.9 | -0.5±2.3 | 0.2 (-0.1, 0.4) | 0.162 |
| | FVS | 8.8±3.7 | -1.0±4.3 | 8.6±2.9 | -0.7±3.3 | 0.3 (0, 0.7) | 0.086 |
| Overweight & obesity | DDS9 | 4.1±1.1 | -0.2±1.4 | 4.3±1.1 | -0.2±1.3 | 0 (0, 0.1) | 0.525 |
| | DDS28 | 6.4±2.1 | -0.4±2.7 | 6.7±1.9 | -0.4±2.2 | 0 (-0.1, 0.1) | 0.836 |
| | FVS | 8.1±3.5 | -0.7±4.3 | 8.4±3.0 | -0.6±3.3 | 0.1(-0.1, 0.2) | 0.608 |
| **For single meal** | | | | | | | |
| Breakfast | DDS9 | 1.8±0.7 | -0.1±0.9 | 1.9±0.7 | 0±0.8 | 0.1 (0, 0.1) | <0.001 |
| | DDS28 | 2.2±0.8 | -0.1±1.1 | 2.2±0.8 | -0.1±0.9 | 0 (0, 0.1) | 0.168 |
| | FVS | 2.3±0.9 | -0.2±1.2 | 2.3±0.8 | -0.1±1.0 | 0.1 (0, 0.1) | 0.067 |
| Lunch | DDS9 | 2.1±0.7 | -0.1±0.9 | 2.1±0.7 | 0±0.9 | -0.1 (-0.1,0) | 0.045 |
| | DDS28 | 2.4±1 | -0.2±1.2 | 2.4±0.9 | -0.1±1.1 | -0.1 (-0.2, 0) | 0.022 |
| | FVS | 2.6±1.1 | -0.3±1.4 | 2.5±1 | -0.1±1.2 | -0.1 (-0.2, 0) | 0.007 |
| Supper | DDS9 | 2.8±0.9 | -0.1±1.1 | 2.8±0.7 | -0.1±0.9 | 0 (0, 0.1) | 0.162 |
| | DDS28 | 3.5±1.3 | -0.2±1.7 | 3.5±1.1 | -0.3±1.4 | 0 (0, 0.1) | 0.378 |
| | FVS | 3.9±1.8 | -0.3±2.2 | 3.8±1.4 | -0.3±1.6 | 0.1 (0, 0.2) | 0.118 |

CI, Confidence Interval; CNP, comprehensive intervention including nutrition education and physical activity; DDS, Dietary Diversity Score; DDS9: Dietary Diversity Score for 9 food groups; DDS28: Dietary Diversity Score for 28 food groups; FVS, Food Variety Score.

A linear mixed model was used for comparison, the effect was adjusted for weight status, sex, age, parental education level and family income level.

## Discussion

This study indicated that the CNP for childhood obesity had no significant effect on either the dietary diversity or the food variety per day, but the significant effects at breakfast and on some foods' consumption (such as cereals and fruits) appeared in 5 cities. For the separate childhood obesity intervention in Beijing, a significant decreasing effect appeared on dietary diversity and food variety in the physical activity group. By analyzing different food consumption, the proportions of children increased more for fruits and decreased less for cereals in the CNP group than the control group after the intervention. The cereals and fruits were the important energy and micronutrient sources, so these findings revealed that these consumption behavior changes may lead to the improvement of diet quality in comprehensive intervention groups. The breakfast is generally taken as the most important meal of the day and is purported to confer a number of benefits for diet quality, health and academic performance. Study reveals that children who habitually consume high quality breakfast are more likely to

**Table 5. The outcomes of the intervention on DDS and FVS in NE or PA intervention study (mean ± SD).**

| Group | Variable | Control group | | NE group | | | | PA group | | | |
|---|---|---|---|---|---|---|---|---|---|---|---|
| | | Baseline | Change | Baseline | Change | Effect (95% CI) | P-value | Baseline | Change | Effect (95% CI) | P-value |
| **For one day** | | | | | | | | | | | |
| Total | DDS9 | 4.0±1.2 | 0.1±1.5 | 4.3±1.1 | 0.1±1.1 | 0.1 (-0.1, 0.3) | 0.294 | 4.1±1.0 | -0.2±1.2 | -0.1 (-0.2, 0) | 0.037 |
| | DDS28 | 5.8±2.1 | 0.6±2.8 | 6.9±1.9 | 0.1±1.9 | 0.1 (-0.3, 0.4) | 0.747 | 6.2±1.7 | 0±1.9 | -0.3 (-0.4, -0.1) | 0.005 |
| | FVS | 6.9±3.1 | 0.7±4.1 | 8.6±3.1 | 0.2±2.8 | 0.4 (-0.2, 0.9) | 0.201 | 7.6±2.3 | -0.2±2.6 | -0.3 (-0.6, 0) | 0.021 |
| Boys | DDS9 | 3.9±1.2 | 0.1±1.5 | 4.3±1.0 | -0.1±1.2 | 0.1 (-0.2, 0.4) | 0.381 | 4.0±1.0 | -0.2±1.2 | -0.1 (-0.3, 0) | 0.147 |
| | DDS28 | 5.7±2.2 | 0.4±2.8 | 6.9±1.9 | -0.2±2.0 | 0.1(-0.4, 0.6) | 0.64 | 5.9±1.7 | -0.2±1.8 | -0.2 (-0.5, 0) | 0.103 |
| | FVS | 6.8±3.1 | 0.5±3.8 | 8.6±3.2 | -0.3±3.1 | 0.5 (-0.2, 1.3) | 0.162 | 7.2±2.3 | -0.5±2.5 | -0.2 (-0.6, 0.1) | 0.187 |
| Girls | DDS9 | 4.2±1.1 | 0.2±1.4 | 4.3±1.1 | 0.1±1.1 | 0.1 (-0.2, 0.4) | 0.431 | 4.3±1.0 | -0.1±1.2 | -0.1 (-0.2, 0.1) | 0.218 |
| | DDS28 | 6.0±2.0 | 0.8±2.8 | 6.9±1.9 | 0.3±1.8 | 0 (-0.5, 0.6) | 0.892 | 6.4±1.7 | 0.1±1.9 | -0.3 (-0.5, 0) | 0.042 |
| | FVS | 7.1±3.1 | 1.0±4.4 | 8.6±3.0 | 0.5±2.5 | 0.3 (-0.6, 1.1) | 0.527 | 7.9±2.3 | 0±2.7 | -0.4 (-0.8, 0) | 0.081 |
| Normal weight & thinness | DDS9 | 4±1.2 | 0.2±1.5 | 4.3±1.0 | 0.1±1.1 | 0.4 (-0.2, 1.0) | 0.162 | 4.1±1.0 | -0.1±1.2 | 0.1 (-0.1, 0.4) | 0.286 |
| | DDS28 | 5.9±2.2 | 0.7±2.9 | 6.9±1.9 | 0.1±1.9 | 1.0 (0, 2.0) | 0.057 | 6.1±1.7 | -0.1±1.8 | 0.4 (-0.1, 0.9) | 0.093 |
| | FVS | 7.0±3.3 | 0.8±4.3 | 8.6±3.1 | 0.2±2.8 | 1.6 (0.1, 3.2) | 0.041 | 7.5±2.4 | -0.3±2.5 | 0.5 (-0.2, 1.2) | 0.135 |
| Overweight & obesity | DDS9 | 4.2±0.9 | -0.3±1.3 | 4.4±1.1 | -0.2±1.3 | 0.1 (-0.1, 0.3) | 0.519 | 4.4±1.1 | -0.2±1.4 | -0.2 (-0.3,0) | 0.006 |
| | DDS28 | 5.6±1.2 | 0±1.9 | 6.8±2.0 | -0.2±2.0 | -0.1 (-0.5, 0.3) | 0.719 | 6.3±1.7 | 0.1±2.3 | -0.4 (-0.6, -0.2) | 0 |
| | FVS | 6.5±1.6 | -0.2±2.5 | 8.7±2.9 | -0.2±2.9 | 0.2 (-0.4, 0.8) | 0.569 | 7.8±2.3 | -0.2±3.0 | -0.5 (-0.8, -0.2) | < 0.001 |
| **For single meal** | | | | | | | | | | | |
| Breakfast | DDS9 | 1.7±0.7 | 0.2±1.0 | 2.1±0.7 | 0.1±0.9 | 0.1 (-0.1, 0.3) | 0.44 | 1.9±0.7 | 0.1±1.0 | 0 (-0.1, 0.1) | 0.754 |
| | DDS28 | 2.1±0.9 | 0.3±1.2 | 2.7±0.8 | 0.1±1.0 | 0.2 (0, 0.4) | 0.126 | 2.2±0.7 | 0.1±1.1 | 0 (-0.1, 0.1) | 0.903 |
| | FVS | 2.1±0.9 | 0.3±1.3 | 2.8±1.0 | 0.1±1.0 | 0.2 (-0.1, 0.4) | 0.155 | 2.3±0.8 | 0.1±1.1 | 0 (-0.1, 0.1) | 0.973 |
| Lunch | DDS9 | 2.1±0.8 | 0.2±1.1 | 2.4±0.7 | 0±0.9 | 0.3 (0.2, 0.5) | 0 | 2.2±0.8 | 0.1±1.1 | 0 (-0.1, 0.1) | 0.94 |
| | DDS28 | 2.3±1 | 0.3±1.5 | 2.9±0.9 | 0±1 | 0.3 (0, 0.5) | 0.03 | 2.5±0.9 | 0.2±1.3 | -0.1 (-0.2, 0) | 0.082 |
| | FVS | 2.4±1.1 | 0.4±1.6 | 3.1±1.1 | -0.1±1.1 | 0.5 (0.2, 0.8) | 0.001 | 2.5±0.9 | 0.2±1.3 | -0.1 (-0.2, 0.1) | 0.197 |
| Supper | DDS9 | 2.5±0.9 | 0.1±1.3 | 2.7±1 | 0.2±1 | 0.2 (0, 0.4) | 0.029 | 2.5±0.8 | 0±1 | -0.1 (-0.2, 0) | 0.046 |
| | DDS28 | 3±1.2 | 0.2±1.6 | 3.3±1.2 | 0.2±1.3 | 0.3 (0, 0.5) | 0.018 | 3.1±1 | -0.1±1.2 | -0.1 (-0.3, 0) | 0.023 |
| | FVS | 3.2±1.5 | 0.2±2.1 | 3.7±1.6 | 0.3±1.7 | 0.3 (0,0.6) | 0.054 | 3.4±1.2 | -0.1±1.5 | -0.2 (-0.3, 0) | 0.017 |

CI, Confidence Interval; DDS, Dietary Diversity Score; DDS9: Dietary Diversity Score for 9 food groups; DDS28: Dietary Diversity Score for 28 food groups; FVS, Food Variety Score; NE, nutrition education intervention; PA, physical activity intervention.

A linear mixed model was used for comparison, the effect was adjusted for weight status, sex, age, parental education level and family income level.

have better nutrient intake [30]. So, the improvement of dietary diversity and food variety at breakfast could increase the diet quality and benefit the children.

Much evidence indicates that physical activity can considerably influence childhood growth [31, 32], while nutrition education provides children with necessary dietary knowledge. There's also study finding that encouraging physical activity could decrease the likelihood of choosing healthy food [33]. In PA group, the significant decreasing effects on dietary diversity and food variety appeared and some healthy food consumption proportions decreased, such as of vegetables, fruits, fungi and algae, nuts and seeds, and fish and shellfish. Most people who have not received nutrition training may intuitively choose to decrease their food intake rather than adjusting to healthy dietary patterns for obesity prevention, such as reducing high-energy food and increasing fruits and vegetables. This may lead the children who only undergo PA to reduce the attention to diet. It implied that the side effect of PA on diet should be noticed in the future program.

It has been demonstrated that both nutrition education and lifestyle intervention could improve the dietary diversity among children or adults [15, 34]. An analysis of over 300 studies

**Table 6. The proportion of children consuming different food groups in CNP study.**

| Food | Control group | | CNP group | | | |
|---|---|---|---|---|---|---|
| | **Baseline (n (%))** | **Change (n (%))** | **Baseline (n (%))** | **Change (n (%))** | **Intervention effect (OR, 95% CI)** | **P-value** |
| Cereals (>3 times /3 days) | 1530 (79.9) | -206 (-10.8) | 1690 (78.5) | -117 (-5.4) | 1.4 (1.1, 1.7) | 0.009 |
| Meat (>3 times /3 days) | 1130 (59.0) | -14 (-0.8) | 1345 (62.5) | 65 (3.0) | 1.2 (1.0, 1.5) | 0.091 |
| Vegetables (>3 times /3 days) | 1259 (65.7) | -8 (-0.4) | 1473 (68.4) | -35 (-1.6) | 0.9 (0.7, 1.1) | 0.521 |
| Fruits (≥1 time /3 days) | 881 (46.0) | -37 (-1.9) | 969 (45.0) | 108 (5.0) | 1.4 (1.1, 1.7) | 0.003 |
| Dairy (≥1 time /3 days) | 1318 (68.8) | -86 (-4.5) | 1536 (71.4) | -85 (-4.0) | 1.0 (0.8, 1.2) | 0.832 |
| Eggs (≥1 time /3 days) | 1424 (74.3) | 81 (4.2) | 1688 (78.4) | -3 (-0.1) | 0.8 (0.7, 1.0) | 0.031 |
| Fish and shellfish (≥1 time /3 days) | 1039 (54.2) | -48 (-2.5) | 1231 (57.1) | -181 (-8.4) | 0.7 (0.6, 0.9) | 0.005 |
| Fungi and algae (≥1 time /3 days) | 676 (35.3) | -10 (-0.5) | 758 (35.2) | -14 (-0.6) | 1.0 (1.2, 1.2) | 0.702 |
| Dried legumes (≥1 time /3 days) | 1032 (53.9) | 48 (2.5) | 1270 (59.0) | -179 (-8.3) | 0.6 (0.5, 0.7) | 0.001 |
| Nuts and seeds (≥1 time /3 days) | 312 (16.3) | -64 (-3.4) | 265 (12.3) | -59 (-2.7) | 0.9 (0.7, 1.2) | 0.665 |
| Snack (≥1 time /3 days) | 352 (18.4) | -39 (-2.1) | 432 (20.1) | -90 (-4.2) | 0.8 (0.7, 1.1) | 0.162 |
| Fast foods (≥1 time /3 days) | 1349 (70.5) | -109 (-5.8) | 1510 (70.1) | -120 (-5.6) | 1.0 (0.9, 1.2) | 0.841 |
| Beverages (≥1 time /3 days) | 665 (34.7) | -202 (-10.5) | 672 (31.2) | -195 (-9.1) | 1.1 (0.9, 1.4) | 0.561 |
| Sugars and preserves (≥1 time /3 days) | 450 (23.5) | -66 (-3.5) | 445 (20.7) | -64 (-3.0) | 1.0 (0.8, 1.2) | 0.839 |

CI, Confidence Interval; CNP, comprehensive intervention including nutrition education and physical activity; OR, Odds Ratio.

A generalized linear mixed model was used for comparison, the effect was adjusted for sex, age, parental education level and family income level.

shows that nutrition education is more likely to be effective when it focuses on behavior/action (rather than knowledge only) and systematically links relevant theory, research and practice [35–38]. The estimated probabilities for obesity were 13.5 percentage points higher among children who consumed a healthy diet but were physically inactive and 3.1 percentage points higher among children who consumed an unhealthy diet but were physically active compared with those who consumed a healthy diet and physically active [39]. This result indicates that PA is much more likely to have a decreasing effect on BMI compared with NE alone. Although the effect of NE intervention on weight reduction was not as obvious in the short term, it is important to help children shape their health diet habits for the long run. Previous research has demonstrated that multicomponent programs involving parents aiming at food sources both in and outside of school and focusing on a variety of unhealthy food items seem less likely to fail [40]. The parents and canteen staff also play important roles in children's eating habits and food choices. The importance of participatory approaches and parental support for school-based health promotion have been demonstrated by many studies. The qualitative research data showed that low parental response could aggravate their children's unhealthy eating behavior [41–44]. Therefore, both of them were covered in our program.

China is experiencing an accelerated process of nutritional transition. During this process, very rapid changes, such as the increase in obesity and obesity-related chronic diseases, have occurred. Studies have identified that these changes are due to a broader dietary offering, changes in eating patterns and a considerable increase in sedentary behaviors/lifestyle [45, 46]. To resolve this problem, some school-based programs for childhood obesity prevention were implemented to develop a strategy of obesity prevention and control. School-based interventions have been clearly shown to be more effective than interventions in other settings [47, 48]. The school-based nutrition education has more extrusive advantages for students than community-based nutrition education, such as offering interacting with educators delivering the intervention and positive school infrastructure and environment. The frequent interactions of policies, curricula and personnel also increases the familiarity with and use of nutrition

**Table 7. The proportion of children consuming different food groups in NE or PA study (%).**

| Food | Control group | | NE group | | | | PA group | | | |
|---|---|---|---|---|---|---|---|---|---|---|
| | Baseline (n (%)) | Change (n (%)) | Baseline (n (%)) | Change (n (%)) | Intervention effect (OR, 95% CI) | P | Baseline (n (%)) | Change (n (%)) | Intervention effect (OR, 95% CI) | P-value |
| Cereals (>3 times /3 days) | 88 (72.1) | 7 (5.8) | 301 (85.5) | 20 (5.7) | 0.8 (0.3, 1.2) | 0.634 | 258 (85.1) | 6 (2.0) | 0.8 (0.3, 1.2) | 0.650 |
| Meat (>3 times /3 days) | 42 (34.4) | -5 (-4.1) | 136 (38.6) | 31 (8.8) | 1.5 (0.8,3) | 0.199 | 86 (28.4) | 4 (1.3) | 1.2 (0.6, 2.5) | 0.557 |
| Vegetables (>3 times /3 days) | 63 (51.6) | -8 (-6.5) | 238 (67.6) | 48 (13.7) | 2.2 (1.2, 4.5) | 0.014 | 224 (73.9) | -50 (-16.5) | 0.6 (0.3, 1.0) | 0.061 |
| Fruits (≥1 time /3 days) | 44 (36.1) | 8 (6.5) | 208 (59.1) | -71 (-20.2) | 0.3 (0.2, 0.6) | 0.001 | 149 (49.2) | -31 (-10.3) | 0.5 (0.3, 0.9) | 0.023 |
| Dairy (≥1 time /3 days) | 96 (78.7) | -17 (-13.9) | 262 (74.4) | -20 (-5.6) | 1.4 (0.7, 2.5) | 0.459 | 210 (69.3) | -35 (-11.5) | 1.0 (0.6,2.2) | 0.837 |
| Eggs (≥1 time /3 days) | 93 (76.2) | 1 (0.8) | 328 (93.2) | -9 (-2.6) | 0.5 (4.5, 1.2) | 0.123 | 245 (80.9) | 9 (2.9) | 0.9 (2.2, 2.0) | 0.865 |
| Fish and shellfish (≥1 time /3 days) | 22 (18.0) | 3 (2.5) | 88 (25.3) | -5 (-1.7) | 0.8 (0.4, 1.7) | 0.527 | 64 (21.1) | -10 (-3.3) | 0.7 (0.3, 1.4) | 0.260 |
| Fungi and algae (≥1 time /3 days) | 22 (18.0) | 4 (3.3) | 120 (34.1) | 90 (25.6) | 2.5 (1.1, 5.0) | 0.024 | 81 (26.7) | -2 (-0.6) | 0.7 (0.3, 1.4) | 0.275 |
| Dried legumes (≥1 time /3 days) | 47 (38.5) | 23 (18.9) | 213 (60.5) | 7 (2.0) | 0.5 (0.2, 0.8) | 0.011 | 157 (51.8) | 16 (5.3) | 0.6 (0.3, 1.0) | 0.048 |
| Nuts and seeds (≥1 time /3 days) | 13 (10.7) | -4 (-3.3) | 60 (17.0) | 19 (5.4) | 2.2 (1.2, 6.1) | 0.103 | 51 (16.8) | -13 (-4.3) | 0.9 (0.3, 2.5) | 0.875 |
| Snack (≥1 time /3 days) | 19 (15.6) | -10 (-8.2) | 151 (42.9) | -108 (-30.7) | 0.3 (0.1, 0.7) | 0.006 | 65 (21.5) | -41 (-13.6) | 0.7 (0.3, 1.8) | 0.44 |
| Fast foods (≥1 time /3 days) | 71 (58.2) | 11 (9.0) | 248 (70.5) | -46 (-13.1) | 0.4 (0.2, 0.7) | 0.002 | 172 (56.8) | -22 (-7.3) | 0.5 (0.3, 0.9) | 0.027 |
| Beverages (≥1 time /3 days) | 31 (25.4) | -2 (-1.6) | 147 (41.8) | -25 (-7.1) | 0.9 (0.5, 1.8) | 0.697 | 111 (36.6) | -22 (-7.2) | 0.8 (0.4, 1.7) | 0.641 |
| Sugars and preserves (≥1 time /3 days) | 16 (13.1) | 6 (4.9) | 53 (15.1) | 1 (0.2) | 0.7 (0.3, 1.5) | 0.309 | 40 (13.2) | -2 (-0.7) | 0.6 (0.2, 1.4) | 0.189 |

CI, Confidence Interval; NE, nutrition education intervention; OR, Odds Ratio; PA, physical activity intervention.

A generalized linear mixed model was used for comparison, the effect was adjusted for sex, age, parental education level and family income level.

knowledge among students [49–51]. To promote nutrition education, some developed countries have developed strategies for school-based nutrition education. The state-level school nutrition coordinators setting reflects the greater historical emphasis on school nutrition and food service policies than on physical activity in the USA, and more types of implementation support to schools for nutrition and food service than for physical activity could be provided by state agencies [52]. However, in China, nutrition education has not attracted enough attention from schools before [53]. With recent the advancement of the State Council's Action Plan for Healthy China, nutrition and health education for children has also been included in the action plan [54]. This will help to promote widespread nutrition education in school.

The main limitation of this study is that the measurements were obtained during only one school year, without evaluation of the long-term impact. Tracking the children for another year would have been very difficult in practice, and students in grade 5 graduated from primary schools and attended different middle schools. For such large-scale follow-up in six cities, much more expense would be needed. Another limitation of our study was the intervention in urban areas but not in rural areas. The prevalence of childhood obesity in rural areas has been increasing more quickly, so the nutrition education is equally important in rural

areas. In particular, children in rural areas of China have access to many unhealthy snack foods in recent years. Additionally, some students always had lunch in private canteens outside of school, but our intervention could not cover such canteens, which may have influenced the intervention effect on the diet. At last, the subgroup analysis results should be treated cautiously, because the subgroup was not considered in the sample design.

## Conclusions

These results indicated that though CNP had no significant effect on overall dietary diversity and food variety per day, the significant effects were shown on some foods' consumption, and the dietary diversity and food variety increases appeared at breakfast significantly. Children exposed only to the physical activity intervention or nutrition education intervention as an obesity intervention didn't appear the positive effect on dietary diversity either. We should pursue a comprehensive intervention approach to changing school policies and practices that addresses both nutrition education and physical activity over time.

## Supporting information

**S1 Data.**
(XLSX)

**S1 Checklist. CONSORT checklist.**
(DOCX)

**S1 File. Study protocol (main points).**
(DOC)

**S2 File.**
(PDF)

## Acknowledgments

We would like to acknowledge the support from all the team members and the participating students, teachers, parents and local education and health staff.

## Author Contributions

**Conceptualization:** Haiquan Xu, Olivier Ecker, Kevin Chen, Guansheng Ma.

**Data curation:** Haiquan Xu, Qian Zhang, Songming Du, Ailing Liu, Yanping Li.

**Formal analysis:** Haiquan Xu, Olivier Ecker.

**Funding acquisition:** Haiquan Xu, Guansheng Ma.

**Investigation:** Haiquan Xu, Qian Zhang, Songming Du, Ailing Liu, Yanping Li, Xiaoqi Hu, Tingyu Li, Hongwei Guo, Ying Li, Guifa Xu, Weijia Liu, Jun Ma, Guansheng Ma.

**Methodology:** Haiquan Xu, Olivier Ecker, Kevin Chen, Guansheng Ma.

**Project administration:** Guansheng Ma.

**Resources:** Haiquan Xu, Guansheng Ma.

**Software:** Haiquan Xu.

**Supervision:** Kevin Chen, Guansheng Ma.

**Writing – original draft:** Haiquan Xu, Olivier Ecker.

**Writing – review & editing:** Haiquan Xu, Olivier Ecker, Qian Zhang, Songming Du, Ailing Liu, Yanping Li, Xiaoqi Hu, Tingyu Li, Hongwei Guo, Ying Li, Guifa Xu, Weijia Liu, Jun Ma, Junmao Sun, Kevin Chen, Guansheng Ma.

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
