## [Decision Letter · Decision Letter 0]

27 Oct 2019

PONE-D-19-24892

Childhood obesity interventions only focusing on physical activity decreases dietary diversity among younger children: Evidence from a school-based randomized controlled trial in China

PLOS ONE

Dear Dr. Xu,

Thank you for submitting your manuscript to PLOS ONE. After careful consideration, we feel that it has merit but does not fully meet PLOS ONE’s publication criteria as it currently stands. Therefore, we invite you to submit a revised version of the manuscript that addresses the points raised during the review process.

We would appreciate receiving your revised manuscript by Dec 11 2019 11:59PM. To enhance the reproducibility of your results, we recommend that if applicable you deposit your laboratory protocols in protocols.io, where a protocol can be assigned its own identifier (DOI) such that it can be cited independently in the future. For instructions see: http://journals.plos.org/plosone/s/submission-guidelines#loc-laboratory-protocols

We look forward to receiving your revised manuscript.

Kind regards,

Mahdieh Abbasalizad Farhangi

Academic Editor

PLOS ONE

Journal Requirements:

3. We noticed you have some minor occurrence(s) of overlapping text with the following previous publication(s), which needs to be addressed:

https://doi.org/10.1186/s12991-017-0162-2

https://doi.org/10.6133/apjcn.112016.05

https://doi.org/10.1371/journal.pone.0043183

https://doi.org/10.1186/s12937-017-0299-5

https://doi.org/10.1186/s12889-016-3878-z

http://dx.doi.org/10.5888/pcd13.160032

In your revision ensure you cite all your sources (including your own works), and quote or rephrase any duplicated text outside the Methods section. Further consideration is dependent on these concerns being addressed.

Reviewers' comments:

Reviewer's Responses to Questions

**Comments to the Author**

1. Is the manuscript technically sound, and do the data support the conclusions?

Reviewer #1: Partly

Reviewer #2: Partly

Reviewer #3: Yes

2. Has the statistical analysis been performed appropriately and rigorously? 

Reviewer #1: No

Reviewer #2: No

Reviewer #3: Yes

3. Have the authors made all data underlying the findings in their manuscript fully available?

Reviewer #1: No

Reviewer #2: No

Reviewer #3: Yes

4. Is the manuscript presented in an intelligible fashion and written in standard English?

Reviewer #1: Yes

Reviewer #2: Yes

Reviewer #3: Yes

5. Review Comments to the Author

Reviewer #1: Childhood obesity interventions only focusing on physical activity decreases dietary

diversity among younger children: Evidence from a school-based randomized

controlled trial in China

The title of study highlighted that focusing on physical activity decrease dietary diversity among children, while the evidences provided in the introduction, method, result and discussion weren’t enough to convince the readers. I am interested to know how Pa interventions decrease DDS but unfortunately couldn’t find strong procedure to prove it.

Method

The intervention elements are unclear for all comprehensive, nutrition education, and PA. Please, clarify the interventions program in details.

PA interventions exist but I can’t find any information about PA measurements in method and results.

Reviewer #2: This paper reported a secondary analysis of the multicentre cluster-randomized controlled trial conducted in China between 2009 and 2010 that aimed to evaluate the effects of a comprehensive school-based intervention on childhood obesity prevention. The main trial was designed into two separate studies, conducted in Beijing and other 5 cities separately. The Beijing study involved 8 schools with 3 schools randomly allocated to receive nutrition education (NE) only, 3 school to receive physical activity (PA) only, and 2 control schools with no intervention. The other study recruited 6 schools in each of the 5 cities (30 schools in total), with 3 schools randomly allocated to receive a comprehensive intervention including both NE and PA (CNP), and 3 control schools with no intervention. The programme was implemented for 2 semesters over one year, and data were collected at baseline and at the end of the study. Among the 38 primary schools included in the trial, two classes were selected from each grade (1-5) per school, but only grades 2-4 were designed to collect the dietary records using 24-h dietary recall for 3 consecutive days. The current study aimed to explore the effects of NE and PA on dietary diversity compared to the control groups, measured using dietary diversity scores (DDS9 and DDS28) and food variety score (FVS). The proportions of different foods consumed were also calculated.

The manuscript was well written, and the study design was clearly documented. My major comments are on the statistical analysis and results reported in this paper, which should follow the CONSORT 2010 guidelines with extension to cluster randomised trials.

As a cluster randomised trial conducted separately in different cities in China, it is important to provide more information on the schools recruited to this study and how they were allocated to each group. Was this decided pragmatically or allocated using random numbers? What factors were considered in the selection of schools and were they stratified in randomisation?

As shown in Table 1, the CNP group (15 schools in 5 cities) received all the NE and PA interventions that was implemented separately in 3 schools in Beijing. With a much larger sample size and comprehensive interventions, the results didn’t show significant effects on the outcomes. In comparison, both NE and PA groups in Beijing schools showed some effects at the end of the trial, especially in the PA group compared to the 2 control schools. Were any process evaluations conducted during the trial that could assess the delivery of different intervention components and the level of protocol compliance?

Although the present analysis on dietary diversity was not the primary outcome of the trial, it would still be useful to prioritize the outcomes considered in this study and define the minimal effect sizes so that both clinical and statistical significances could be established and used in interpretation.

To clarify, were the outcomes defined at the end of study or as the change from baseline? Were the subgroup analyses by gender and weight status pre-defined or determined post-hoc? What was the rationale to evaluate the outcomes for breakfast separately? They should be described in the statistical analysis section.

For randomised controlled trials, it is not recommended to include baseline outcome measured prior to randomisation as part of the outcome variable assessed post randomisation. As suggested in the EMA guideline on adjustment for baseline covariates in clinical trials, the baseline value of a continuous outcome measure should usually be included as a covariate. This applies whether the outcome variable is defined as the ‘raw outcome’ or as the ‘change from baseline’. In current statistical analysis models, however, a time variable was added with 0 = baseline and 1 = the end of study on the outcome measure with an interaction term between time and treatment group. What terms were used to estimate the intervention effects reported in the table? A more accurate regression model is to fit the outcome measured at the end of the study, with the baseline value and other pre-defined confounders added to the model as covariates for adjustment. The intervention effect is then estimated using the model coefficient for treatment group.

As school was the randomisation unit rather than school class, what was the rationale to fit class as the random effect only?

Please follow the CONSORT 2010 statement with extension to cluster randomised trials in reporting figures and tables. The flow diagram should indicate the follow up schedule, and report both numbers of schools and students in each group. Baseline table should not report p-values, and include both school- and individual-level data collected. The result tables are hard to read with too much information in one table. I would suggest reporting the CNP study and NE/PA study in separate tables, with the randomised groups reported in columns and the outcome measures in rows. Intra-cluster correlation coefficient needs to be reported on each outcome, which is strongly recommended in cluster randomised trials. For the trial conducted in Beijing that included three randomised arms (NE, PA, control), the results for each outcome should be reported together as a multi-arm study rather than two separate analyses for NE vs control and PA vs control.

With numerous statistical tests conducted overall and by subgroup, the results must be interpreted with caution as the Type 1 error rate would be inflated without adjustment for multiple comparisons.

Based on the results from 3 intervention schools and 2 control schools in Beijing, the title and conclusions focusing only on the significant effects of PA on dietary diversity seem too strong. For example on page 17, line 270, it was reported that the effects on DDS9, DDS28 and FVS were -0.12 (95% CI: -0.38, 0.14; p = 0.365), -0.61 (95% CI: -1.04, -0.17; p = 0.007) and -0.71 (95% CI: -1.35, -0.08; p = 0.028) in the NE group and -0.34 (95% CI: -0.61, -0.08; p = 0.011), -0.73 (95% CI: -1.18, -0.28; p = 0.001) and -1.1 (95% CI: -1.74, -0.45; p = 0.001) in the PA group, respectively. On page 26, line 400, the authors concluded that this result indicated that CNP and NE had no significant effect on the dietary diversity but the dietary diversity decreased in children exposed only to the physical activity intervention. Was this based on DDS9 which was one of the study outcomes? As mentioned earlier, it would be useful to prioritize the outcomes and define the minimal effect sizes so that both clinical and statistical significances could be established and used in interpretation.

Reviewer #3: Childhood obesity and overweight are public health problems and represent an important issue in china population. In this regard different interventions (dietary and PA interventions) are designed for children in schools. The study aimed to understand the association of the type of intervention and dietary diversity score among young children. This is an important topic that could be used for planning a proper prevention program in this age group.

The article is well written however, it presents some critical points:

1. In the introduction: please add some studies that assess the effect of education or PA on diet quality or diversity.

2. from the introduction section, it is not inferred that why the authors assume that the type of obesity programs could affect on DDS differently.

3. in the method section, please add the method of randomization.

4. from the method, it is not obvious that how these 38 schools were selected.

5. it is better to add flow chart for schools selections and in this indicate how many school were invited to take part in these programs and how many accepted it.

6. IN TABLES: Please the statistical analysis used for comparisons or associations as a table footnote.

7. Table 3, 4: it seems that the data presented in table 3 and 4 could be merged, so the comparisons become more understandable.

8. Table 5, 6: it seems that the data presented in table 5 and 6 could be merged, so the comparisons become more understandable

9. Why the authors analysed the data secretively for breakfast?

10. the second paragraph of discussion is not related to this part. it may be more suitable for introduction section.

11. in the discussion section it is recommended to discuss the finding of the present study. In this part, the author mostly discussed about obesity, not diet quality and diversity.

6. PLOS authors have the option to publish the peer review history of their article (what does this mean?). If published, this will include your full peer review and any attached files.

Reviewer #1: No

Reviewer #2: No

Reviewer #3: No

---

## [Author Response · Author response to Decision Letter 0]

30 Dec 2019

appreciate for the comments of reviewers and editor

---

## [Decision Letter · Decision Letter 1]

23 Apr 2020

PONE-D-19-24892R1

The effect of comprehensive intervention for childhood obesity on dietary diversity among younger children: Evidence from a school-based randomized controlled trial in China

PLOS ONE

Dear Dr. Xu,

Thank you for submitting your manuscript to PLOS ONE. After careful consideration, we feel that it has merit but does not fully meet PLOS ONE’s publication criteria as it currently stands. Therefore, we invite you to submit a revised version of the manuscript that addresses the points raised during the review process. In your revised version, please format the tables properly, so that they can be read more easily by reviewers.

We would appreciate receiving your revised manuscript by Jun 07 2020 11:59PM. To enhance the reproducibility of your results, we recommend that if applicable you deposit your laboratory protocols in protocols.io, where a protocol can be assigned its own identifier (DOI) such that it can be cited independently in the future. For instructions see: http://journals.plos.org/plosone/s/submission-guidelines#loc-laboratory-protocols

We look forward to receiving your revised manuscript.

Kind regards,

Seth Adu-Afarwuah

Academic Editor

PLOS ONE

Reviewers' comments:

Reviewer's Responses to Questions

**Comments to the Author**

1. If the authors have adequately addressed your comments raised in a previous round of review and you feel that this manuscript is now acceptable for publication, you may indicate that here to bypass the “Comments to the Author” section, enter your conflict of interest statement in the “Confidential to Editor” section, and submit your "Accept" recommendation.

Reviewer #1: All comments have been addressed

Reviewer #2: All comments have been addressed

Reviewer #3: All comments have been addressed

2. Is the manuscript technically sound, and do the data support the conclusions?

Reviewer #1: Yes

Reviewer #2: Partly

Reviewer #3: Yes

3. Has the statistical analysis been performed appropriately and rigorously? 

Reviewer #1: Yes

Reviewer #2: No

Reviewer #3: Yes

4. Have the authors made all data underlying the findings in their manuscript fully available?

Reviewer #1: Yes

Reviewer #2: No

Reviewer #3: Yes

5. Is the manuscript presented in an intelligible fashion and written in standard English?

Reviewer #1: Yes

Reviewer #2: No

Reviewer #3: Yes

6. Review Comments to the Author

Reviewer #1: (No Response)

Reviewer #2: Thank you for the revised paper and responses to all the comments. With further clarification on the original trial design, my concerns remain on the overall conduct of the two studies and the analysis results reported using the specified models.

As shown in Table 1, the two separate cluster randomised trials conducted in Beijing and other 5 cities used same intervention components on PA and NE (separately or combined). The authors stated that all the schools implemented the programme as proposed. If this was the case, it would be hard to evaluate the intervention effects with observed differences on the demographic characteristics and estimated effect sizes between two studies. There are also large geographic differences between these cities in China, and how similar the selected schools were between groups is unknown. If the CNP study was the fundamental part of the trial, the authors should perhaps focus on this main study only and report the change on both BMI (primary outcome) and dietary diversity in the paper.

For a cluster randomised trial, the unit of randomisation is at the cluster level while the data collection may include both cluster and individual participants’ data. The regression models should consider the within- and between-cluster variations, and fit both fixed and random effects using the mixed models. Fitting baseline outcome value as a covariate in the fixed effect model is applicable to both continuous and categorical outcomes, not just the change from baseline scores. The revised equation (1) was fitted using a standard linear regression model without a random effect. The equation (2) fitted the baseline value as part of the outcome measures in the GLMM model. When an interaction term was added in the logistic model, the beta coefficient for the interaction term should not be interpreted independently from the main terms on treatment and time. The model-adjusted group difference needs to be estimated at the end of the study using the odds ratio (i.e. exponential of the raw model estimate).

Reviewer #3: (No Response)

7. PLOS authors have the option to publish the peer review history of their article (what does this mean?). If published, this will include your full peer review and any attached files.

Reviewer #1: No

Reviewer #2: No

Reviewer #3: No

---

## [Author Response · Author response to Decision Letter 1]

7 May 2020

Dear Dr. Seth Adu-Afarwuah, Editors and Reviewers:

We very much appreciate your thoughtful comments on our manuscript (PONE-D-19-24892R1, “The effect of comprehensive intervention for childhood obesity on dietary diversity among younger children: Evidence from a school-based randomized controlled trial in China”). We have carefully considered each comment and responded point-by-point below. We have made corresponding changes in the manuscript with tracked changes highlighting both additions and deletions. 

In addition, we have carefully checked the format requirement and revised the manuscript according to the requirement of PLOS ONE. 

We believe that your comments and our responses to them have improved our manuscript considerably, and we hope you find our revised manuscript suitable for publication.

---

## [Decision Letter · Decision Letter 2]

1 Jun 2020

PONE-D-19-24892R2

The effect of comprehensive intervention for childhood obesity on dietary diversity among younger children: Evidence from a school-based randomized controlled trial in China

PLOS ONE

Dear Dr. Xu,

Thank you for submitting your manuscript to PLOS ONE. After careful consideration, we feel that it has merit but does not fully meet PLOS ONE’s publication criteria as it currently stands. Therefore, we invite you to submit a revised version of the manuscript that addresses the points raised during the review process.

We look forward to receiving your revised manuscript.

Kind regards,

Seth Adu-Afarwuah

Academic Editor

PLOS ONE

Reviewers' comments:

Reviewer's Responses to Questions

**Comments to the Author**

1. If the authors have adequately addressed your comments raised in a previous round of review and you feel that this manuscript is now acceptable for publication, you may indicate that here to bypass the “Comments to the Author” section, enter your conflict of interest statement in the “Confidential to Editor” section, and submit your "Accept" recommendation.

Reviewer #2: All comments have been addressed

2. Is the manuscript technically sound, and do the data support the conclusions?

Reviewer #2: Yes

3. Has the statistical analysis been performed appropriately and rigorously? 

Reviewer #2: Yes

4. Have the authors made all data underlying the findings in their manuscript fully available?

Reviewer #2: Yes

5. Is the manuscript presented in an intelligible fashion and written in standard English?

Reviewer #2: Yes

6. Review Comments to the Author

Reviewer #2: I would like to suggest the following minor changes to the manuscript.

In statistical analysis section, the authors stated that Pij in equation (2) is the change of probability of an event from baseline to end for child i at level j. I disagree with this statement as the logit link is suitable to a binary outcome (Yes or No), which cannot be calculated as change from baseline for each participant. Using same notations as equation (1), the authors should say that the same linear equation was applied to binary outcomes with a logit link as part of generalised linear models. With no time effect in the model, the sentence "The effect of the intervention was evaluated by testing the interaction term between time and treatment" should be removed.

In the results, the authors have used the difference in change of proportions between two groups to quantify the intervention effect on food group consumption, rather than the estimated odds ratio and 95% CI. Although I understand that the difference in proportions is more intuitive to interpret, the adjusted model estimate is more robust which has taken into account baseline covariates and random cluster effect.

Please follow the CONSORT 2010 Explanation and Elaboration in reporting tables and figures, with the extension to cluster randomised trials.

7. PLOS authors have the option to publish the peer review history of their article (what does this mean?). If published, this will include your full peer review and any attached files.

Reviewer #2: No

---

## [Author Response · Author response to Decision Letter 2]

6 Jun 2020

Dear Dr. Seth Adu-Afarwuah, Editors and Reviewers:

We very much appreciate your thoughtful comments on our manuscript (PONE-D-19-24892R2, “The effect of comprehensive intervention for childhood obesity on dietary diversity among younger children: Evidence from a school-based randomized controlled trial in China”). We have carefully considered each comment and responded point-by-point below. We have made corresponding changes in the manuscript with tracked changes highlighting both additions and deletions. 

In addition, we have carefully checked the format requirement and revised the manuscript according to the requirement of PLOS ONE. 

We believe that your comments and our responses to them have improved our manuscript considerably, and we hope you find our revised manuscript suitable for publication.

---

## [Decision Letter · Decision Letter 3]

26 Jun 2020

The effect of comprehensive intervention for childhood obesity on dietary diversity among younger children: Evidence from a school-based randomized controlled trial in China

PONE-D-19-24892R3

Dear Dr. Xu,

We’re pleased to inform you that your manuscript has been judged scientifically suitable for publication and will be formally accepted for publication once it meets all outstanding technical requirements.

Kind regards,

Seth Adu-Afarwuah

Academic Editor

PLOS ONE

Additional Editor Comments (optional):

Reviewers' comments:

Reviewer's Responses to Questions

**Comments to the Author**

1. If the authors have adequately addressed your comments raised in a previous round of review and you feel that this manuscript is now acceptable for publication, you may indicate that here to bypass the “Comments to the Author” section, enter your conflict of interest statement in the “Confidential to Editor” section, and submit your "Accept" recommendation.

Reviewer #2: All comments have been addressed

2. Is the manuscript technically sound, and do the data support the conclusions?

Reviewer #2: (No Response)

3. Has the statistical analysis been performed appropriately and rigorously? 

Reviewer #2: (No Response)

4. Have the authors made all data underlying the findings in their manuscript fully available?

Reviewer #2: (No Response)

5. Is the manuscript presented in an intelligible fashion and written in standard English?

Reviewer #2: (No Response)

6. Review Comments to the Author

Reviewer #2: (No Response)

7. PLOS authors have the option to publish the peer review history of their article (what does this mean?). If published, this will include your full peer review and any attached files.

Reviewer #2: No

---

## [Editor Report · Acceptance letter]

7 Jul 2020

PONE-D-19-24892R3 

The effect of comprehensive intervention for childhood obesity on dietary diversity among younger children: Evidence from a school-based randomized controlled trial in China 

Dear Dr. Xu:

I'm pleased to inform you that your manuscript has been deemed suitable for publication in PLOS ONE. Congratulations! Your manuscript is now with our production department. 

Kind regards, 

on behalf of

Dr. Seth Adu-Afarwuah 

Academic Editor

PLOS ONE